# Stability of Ophthalmic Atropine Solutions for Child Myopia Control

**DOI:** 10.3390/pharmaceutics12080781

**Published:** 2020-08-17

**Authors:** Baptiste Berton, Philip Chennell, Mouloud Yessaad, Yassine Bouattour, Mireille Jouannet, Mathieu Wasiak, Valérie Sautou

**Affiliations:** 1CHU Clermont-Ferrand, Pôle Pharmacie, F-63003 Clermont-Ferrand, France; baptiste-berton@hotmail.fr (B.B.); myessaad@chu-clermontferrand.fr (M.Y.); mjouannet@chu-clermontferrand.fr (M.J.); mwasiak@chu-clermontferrand.fr (M.W.); 2Université Clermont Auvergne, CHU Clermont-Ferrand, CNRS, SIGMA, ICCF, 63000 Clermont-Ferrand, France; ybouattour@chu-clermontferrand.fr (Y.B.); vsautou@chu-clermontferrand.fr (V.S.)

**Keywords:** atropine, ophthalmic solution, stability, myopia

## Abstract

Myopia is an ophthalmic condition affecting more than 1/5th of the world population, especially children. Low-dose atropine eyedrops have been shown to limit myopia evolution during treatment. However, there are currently no commercial industrial forms available and there is little data published concerning the stability of medications prepared by compounding pharmacies. The objective of this study was to evaluate the stability of two 0.1 mg/mL atropine formulations (with and without antimicrobiobial preservatives) for 6 months in two different low-density polyethylene (LDPE) multidose eyedroppers. Analyses used were the following: visual inspection, turbidity, chromaticity measurements, osmolality and pH measurements, atropine quantification by a stability-indicating liquid chromatography method, breakdown product research, and sterility assay. In an in-use study, atropine quantification was also performed on the drops emitted from the multidose eyedroppers. All tested parameters remained stable during the 6 months period, with atropine concentrations above 94.7% of initial concentration. A breakdown product (tropic acid) did increase slowly over time but remained well below usually admitted concentrations. Atropine concentrations remained stable during the in-use study. Both formulations of 0.1 mg/mL of atropine (with and without antimicrobial preservative) were proved to be physicochemically stable for 6 months at 25 °C when stored in LDPE bottles, with an identical microbial shelf-life.

## 1. Introduction

Myopia (or short-sightedness) is an ophthalmic condition that leads to blurred long-distance vision, generally characterized by a refractive error of −0.5 or −1 diopters. Overall, it has been estimated that currently 1.4 billion people in the world are myopic (22.9% of the population), and crude estimations suggest that, by 2050, there will be 4.7 billion people affected (nearly 50% of world population) [1]. The prevalence of myopia is variable between countries, affecting for example about 30% of young adults in Europe [2], 59% in North America [3], to more than 95% in some student populations of Asian countries [4,5], with onset of the disease occurring during childhood and adolescence. Many risk factors have been suggested or clearly identified, such as time spent performing close-vision activities, such as reading or looking at smart-device screens [6], lack of physical exercise, or exposure to sunlight [3,7], often all linked to higher education rates [8]. If left uncorrected, myopia has been shown to have major consequences on children’s level of education, quality of life, and personal and psychological well-being [9], and its economic impact on society has been estimated at US$244 billion from global potential productivity loss [10]. Several recommendations have been proposed to limit the onset of myopia, like encouraging children spending time outside, limiting close-vision activities, and adapting light conditions [11], but whilst fundamental, they might not be sufficient or easily implemented. Current therapeutic options all have limitations; for example, orthokeratology (reshaping of the cornea by a hard, hydrophilic, gas-permeable contact lens worn during sleep and removed during the day) does not stop disease progression to returning to its previous rate after treatment discontinuation and requires high levels of patient compliance [12]. The use of correction glasses or lenses does not address the root-cause, refractive surgery is only curative, and its cost-effectiveness is still uncertain [13]. Pharmacological treatments have however shown some very promising potential. Of these, mydriatic agents such as atropine or tropicamide gained interest early on [14], with atropine being the most studied. Its biological action has been surmised as involving a complex interplay with receptors on different ocular tissues at multiple levels, leading to a decrease in change in the cycloplegic refraction and axial length elongation [15]. Initially tested at 1% concentration during the ATOM1 trial, atropine eyedrops were proved to be effective in controlling myopic progression but caused important visual side effects resulting from cycloplegia and mydriasis [16]. Since then, several clinical trials have evaluated the safety and efficiency of atropine eye drops at lower concentrations. The ATOM2 trial studied myopia progression in 400 children 2 years after treatment with 0.01%, 0.1%, and 0.5% and found the 0.01% concentration to be the concentration causing the least side effects for comparable efficacy in controlling myopia progression [17,18]. Very recently, the LAMP phase 2 trial report confirmed that concentrations ranging from 0.01% to 0.05% were well tolerated in 383 children after two years of treatment, with patients experiencing rare and mild side effects [19], even if the need for photochomratic glasses was higher than 30% for treated patients. The ATOM1, ATOM2, and LAMP trials were all conducted on Asian patients, and some have highlighted the need for high-quality evidence from European populations on atropine effectiveness in controlling myopia progression [20]. However, a smaller study on European paediatric patients also concluded that 0.01% atropine eye drops slowed the rate of myopia progression whilst retaining a favourable safety profile [21].

Despite all this recent and very favourable clinical data, there is still currently no commercially available low dose formulation of atropine eyedrops. In order to treat patients, hospital and compounding pharmacies could produce the desired ophthalmic solution, but the lack of long-term validated stability data severely limits their conservation period by imposing short expiration dates after preparation [22,23]. Indeed, few studies have been published concerning atropine eyedrops stability, but the analyses were either lacking several tests or suffered from shortcomings concerning breakdown product research [24,25,26]. As single-dose container technology is not readily available to compounding pharmacies, multidose eyedroppers (often in low dose polypropylene) are the most used container and therefore the most studied [27,28,29,30]. However, not all of those devices possess a system allowing their content to be preservative-free, and their contents must therefore be preserved. Because 0.1 mg/mL is the concentration with currently the most data concerning safety and efficacy, the aim of this study was therefore to assess the physicochemical stability and to control the sterility of two 0.1 mg/mL atropine ophthalmic solutions (with and without an antimicrobial conservative) in two different low-density polyethylene (LDPE) multidose eyedroppers (one with a sterility-preserving technology allowing the absence of an antimicrobial preservative in the formulation and the other without such a system in order to allow choice of container depending on stability data) at 25 °C for six months in unopened eyedroppers.

## 2. Materials and Methods

### 2.1. Preparation and Storage of Atropine Solution Formulations

Two different formulations of 0.1 mg/mL (0.01%) atropine ophthalmic solutions were prepared:-atropine solution with an antimicrobial preservative (cetrimide) for use with ethylene oxide sterilized white opaque LDPE squeezable multidose eyedroppers (reference VPLA25B10; Laboratoire CAT^®^, Lorris, France)-atropine solution without the preservative for use within gamma-sterilized white opaque LDPE squeezable multidose eyedropper (reference 10002134) equipped with sterility preserving Novelia^®^ caps (reference 20050772; Nemera, La Verpillère, Cedex France).

The details of each formulation are presented in Table 1. All compounds that were used were of pharmaceutical grade.

The formulations were prepared by dissolving atropine into the 0.9% sodium chloride solution at room temperature under gentle agitation before adding the hydrogenophosphate buffer and, finally, if needed, the preservative (cetrimide). For the purpose of the study, batch size was 1 L for both formulations. Cetrimide at a concentration of 0.01% was chosen because it is a preservative commonly used for the antimicrobial preservation of ophthalmic solutions, with an efficacy similar to benzalkonium chloride [31,32].

The obtained atropine solutions were filtered through a 0.22-µm filter (Stericup^®^ Sterile Vacuum Filtration Systems, Merck Millipore, MC2, Clermont-Ferrand, France) and then sterilely distributed (6 mL per unit, for a maximum filling capacity of 8 mL for both multidose eyedroppers) into the eyedroppers under the laminar airflow of an ISO 4.8 microbiological safety cabinet using a conditioning pump (Repeater pump, Baxter, Guyancourt, France). The solution was distributed into the two different low-density polyethylene (LDPE) eyedroppers.

### 2.2. Study Design

The stability of the 0.1 mg/mL atropine solutions was studied for 180 days at 25 °C in unopened eyedroppers and in simulated use conditions for 6 days.

#### 2.2.1. Stability of 0.1 mg/mL Atropine in Unopened Multidose Eyedroppers

The eyedroppers containing atropine were stored upwards in an ICH Q1B compliant climate chamber (BINDER GmbH, Tuttlingen, Germany) at 25 °C ± 2 °C and 60 ± 5% residual humidity until analysis.

Immediately after preparation (day 0) and at days 8, 15, 30, 90, and 180, five units per kind of eyedropper were submitted to the following analyses: visual inspection, chromaticity analysis, atropine quantification, breakdown products (BPs) research (i.e., looking specifically for products resulting from the degradation of atropine), osmolality, pH, and turbidity. Sterility was also assessed using five units for each kind of eyedropper and storage temperature immediately after preparation and after 60 and 180 days of storage. Initial day 0 analyses were performed immediately after conditioning within 4 h after the end of the preparation of the solutions to have results as representative as possible of initial conditions (least degradation or modification of parameters).

#### 2.2.2. Evaluation of Atropine Concentrations in Eye Drops during Simulated Use

Thirty eyedroppers were subjected to simulated patient use: every day for 6 days, one drop from each eyedroppers was manually emitted (i.e., the drop was squeezed out of the bottle as if to be administered to the eye, but instead of being administered, it was collected for analysis) at room temperature. Atropine quantification was then realized in triplicate from 10 collected and pooled drops. In between use, the bottles were stored vertically at 25 °C.

### 2.3. Analyses Performed on the Atropine Solutions

#### 2.3.1. Visual Inspection

The multidose eyedroppers were emptied into glass test tubes, and the atropine solutions were visually inspected under day light and under polarized white light from an inspection station (LV28, Allen and Co., Liverpool, UK). Aspect and colour of the solutions were noted, and a screening for visible macroparticles, haziness, or gas development was performed.

#### 2.3.2. Chromaticity Analysis

Chromaticity and luminance were measured with a UV-visible spectrophotometer (V670, Jasco^®^, Lisses, France) using the mode Color Diagnosis of the built-in software (Spectra Manager^®^, version 2.12.00). The xyY CIE colorimetric system was used. Chromaticity was presented as a two-dimensionl diagram (x and y axes) representing the whole the colour system independently of luminance. uminance is defined as the visual sensation of luminosity of a surface measured by the ratio of the colour’s luminosity (in cd · cm^−2^) over the luminosity of pure white (reference colour) times 100; its value Y ranges therefore between 0 (no luminosity) and 100 (maximum luminosity).

#### 2.3.3. Atropine Quantification and BPs Research

##### Chemicals and Instrumentation

For each unit, atropine was quantified and BPs were detected using the liquid chromatography (LC) method described by the European Pharmacopeia, Atropine monography [33]. The LC system that was used was a Prominence-I LC2030C 3D with diode array detection (Shimadzu France SAS, Marne La Vallée, France), and the associated software used to record and interpret chromatograms was LabSolutions^®^ version 5.82. The LC separation column used was a C18 Synergi^®^ Fusion-RP 80 (150 × 4.6 mm, 4 µm) with an associated guard column (Phenomenex, Le Pecq, France).

The mobile phase was a gradient mixture of phases A and B. Phase A consisted of an aqueous solution of 3.5 g of sodium dodecyl sulphate (SDS) (CAS 1561-21-3, purity > 99%, Sigma-Aldrich, St. Louis, MO, USA) in 606 mL of a 7 g/L solution of potassium dihydrogen phosphate (CAS 10049-21-5, purity > 99%, Sigma-Aldrich, St. Louis, MO, USA) previously adjusted to pH 3.3 with orthophosphoric acid (20,624.295, purity > 85%, Normapur, Prolabo, Paris, France) and mixed with 320 mL of acetonitrile (34851-2, purity > 99.9%, Honeywell, Charlotte, NC, USA). The final pH of phase A was of 3.9. Phase B consisted of 100% acetonitrile. The gradient used is presented in Table 2. All solvents were of analytical grade.

The flow rate through the column for the analysis was set at 1 mL/min, with column thermo-regulation set to a temperature of 30 °C. The injection volume was of 20 µL. The quantification wavelength was set up at 210 nm. BP detection was realized by screening with a diode array detector (DAD) detector from 190 nm to 800 nm.

##### Method Validation

Linearity was initially verified by preparing one calibration curve daily for three days using five concentrations of atropine (European Pharmacopoeia reference standard Y0000878 (Sigma-Aldrich, MC2, Clermont-Ferrand, France) at 10, 20, 60, 100, and 140 µg/mL, diluted in deionized water. Each calibration curve should have a determination coefficient R^2^ equal or higher than 0.999. Homogeneity of the curves was verified using a Cochran test. ANOVA tests were applied to determine applicability. Each day for three days, six solutions of atropine 0.1 mg/mL were prepared, analysed, and quantified using a calibration curve prepared the same day. To verify the method precision, repeatability was estimated by calculating relative standard deviation (RSD) of intraday analysis and intermediate precision was evaluated using an RSD of inter-days analysis. Both RSDs should be less than 5%. Specificity was assessed by comparing the UV spectra DAD detector. Method accuracy was verified by evaluating the recovery of five theoretical concentrations to experimental values found using mean curve equation, and results should be found within the range of 95–105%. The overall accuracy profile was constructed according to Hubert et al. [34,35,36].

The matrix effect was evaluated by reproducing the previous methodology with the presence of all excipients present in the formulation (including the preservative) and by comparing the calibration curves and intercepts.

Atropine impurities described in European Pharmacopeia (atropine impurity B CRS, atropine for peak identification CRS (containing impurities A, D, E, F, G, and H) and tropic acid R (impurity C)) were identified with the same method. Their retention times were collected for potential identification and quantification during stability studies.

In order to exclude potential interference of degradation products with atropine quantification, atropine 0.1 mg/mL solutions was subjected to the following forced degradation conditions: 0.1, 0.5, and 1 N hydrochloric acid for 150 min at 25 °C; 0.1, 0.5, and 1 N chloride acid for 150 min at 90 °C; 0.1, 0.5, and 1 N sodium hydroxide for 30 min at 25 °C; 15% hydrogen peroxide for 60 min at 60 °C and 90 °C; and 30% hydrogen peroxide for 60 and 180 min at 90 °C. Susceptibility to light was performed 3 times after solution preparation for 180 min and 4 and 8 days using an UVA light in climatic chamber (25 °C).

Tropic acid was quantified at 210 nm using the same method as for atropine quantification in the presence of a phosphate buffer, using a calibration curve ranging from 0.1 to 5.0 µg/mL validated using the same methodology as previously described for the validation of the atropine quantification method.

#### 2.3.4. Osmolality, pH, and Turbidity Measurements

For each unit, pH measurements were made using a SevenMultiTM pH-meter with an InLabTM Micro Pro glass electrode (Mettler-Toledo, Viroflay, France). Measures were preceded and followed by instrument validation using standard buffer solution of pH 4 and pH 7 (HANNAH^®^ Instrument, Tannerries, France). Osmolality was measured for each solution using an osmometer Model 2020 Osmometer^®^ (Advanced instruments Inc., Radiometer, SAS, Neuilly Plaisance, France). Turbidity was measured using a 2100Q Portable Turbidimeter (Hach Lange, Marne La Vallée, France), by pooling the five samples per analysed experimental condition and assay time to obtain the necessary volume for the analysis. The results were expressed in Formazin Nephelometric Units (FNU).

#### 2.3.5. Sterility Assay

Sterility was assessed using the European Pharmacopoeia sterility assay (2.6.1). Multidose eyedroppers were opened under the laminar air flow of an ISO 4.8 microbiological safety cabinet, and the contents were filtered under vacuum using a Nalgene^®^ analytical test filter funnel onto a 47-mm diameter cellulose nitrate membrane with a pore size of 0.45 mm (ref 147-0045, Thermo Scientific, purchased from MC2, Clermont-Ferrand CEDEX, France). The membranes were then rinsed with 500 mL deionized water (Versylene^®^; Fresenius Kabi, France, Louviers, France) and divided into two equal parts. Each individual part was transferred to each of a fluid thioglycolate and soya tripcase medium and incubated at 30–35 °C or 20–25 °C, respectively, for 14 days. The culture medium was then examined for colonies.

### 2.4. Data Analysis—Acceptability Criteria

The stability of diluted atropine solutions was assessed using the following parameters: visual aspect of the solution, turbidity, pH, osmolality, atropine concentration, and presence or absence of BPs.

The study was conducted following methodological guidelines issued by the International Conference on Harmonisation for stability studies [37] and recommendations issued by the French Society of Clinical Pharmacy (SFPC) and by the Evaluation and Research Group on Protection in Controlled Atmosphere [38]. A variation of concentration outside the 90–110% range of initial concentration (including the limits of a 95% confidence interval of the measures) was considered as being a sign of instability. Presence of BPs and the variation of the physicochemical parameters were also considered a sign of atropine instability but were interpreted with regards to quantities found in commercial ophthalmic atropine solution (see Appendix A). The observed solutions must be limpid, of unchanged colour, and clear of visible signs of haziness or precipitation. Since there are no standards that define acceptable pH or osmolality variation, pH measures were considered acceptable if they did not vary by more than one pH unit from the initial value [38], and osmolality results were interpreted considering clinical tolerance of the preparation.

## 3. Results

### 3.1. Atropine Quantification and Breakdown Products (BP) Research

Atropine retention time was of 9.7 ± 0.3 min (Figure 1). The chromatographic method used was found linear for concentrations ranging from 0.5 to 140 µg/mL. Average regression equation was y = 22429.5x−13.6, where x is the atropine concentration (in µg/mL) and y is the surface area of the corresponding chromatogram peak. Interception was not significantly different from zero, and average determination coefficient R^2^ of three calibration curves was 0.99999. No matrix effect was detected.

The relative mean trueness bias coefficients were less than 2.75%, except for the 0.5 µg/mL calibration point, for which it was of 3.75%. Mean repeatability RSD coefficient and mean intermediate precision RSD coefficient were less than 2%. The accuracy profile constructed with the data showed that the limits of 95% confidence interval coefficients were all within 3% of the expected value, except for the 10 µg/mL calibration point, for which the lower range limit was −8.8%. The limit of detection was evaluated at 0.05 µg/mL (signal-to-noise ratio of 3.03 and experimentally confirmed by visual analysis of the chromatograms), and the limit of quantification was fixed at 0.5 µg/mL, even if the signal-to-noise ratio was 46, thus potentially indicating that a lower quantification limit could be reached.

No impurities were visible in the initial atropine solution on the reference chromatogram at 210 nm in Figure 1. All the impurities specified by the European Pharmacopeia were detected and identified in Figure 2. The chromatograms presented show separately different impurities as they come from different European Pharmacopoeia reference solutions and were thus analysed sequentially in order to be able to correctly identify each peak using the relative retention times provided in the atropine monography. They were all visible at 210 nm, which allowed maximum sensitivity. No other impurities were detected at other wavelengths.

A summary of the impurity retention times and relative retention times (relative to atropine) is presented in Table 3.

After forced degradation, BPs were detected with a resolution higher than 1.5 of the atropine peak to all its BPs and particularly in alkaline forced conditions. No BPs were detected when atropine solutions were exposed to UVA light, and atropine concentration did not vary after 8 days of UV-Vis exposure. After 1 h at 15% H_2_O_2_ exposure at 60 °C, no loss of atropine concentration was detected. After 1 h at 90°, a loss of 7.9% of atropine was noticed, without any breakdown products being detected. Chromatogram results are showed in Figure 3.

Detailed results of tropic acid quantification are presented in Appendix A.

### 3.2. Stability of Atropine in Unopened Multidose Eyedroppers

#### 3.2.1. Physical Stability

All samples stayed limpid and uncoloured; chromaticity and luminance were unchanged during the study for both tested kind of eyedroppers; and there was no appearance of any visible particulate matter, haziness, or gas development. Initial turbidity was 0.33 and 0.32 FNU respectively for the atropine formulation with and without preservatives and did not vary by more than 0.6 FNU for the formulation with antimicrobial preservative or 0.32 FNU for the formulation without preservative (Table 4).

#### 3.2.2. Chemical Stability

Evolution of pH and osmolality throughout the study is presented Table 5. Throughout the study, osmolality did not vary by more than 3.7% (15 mOsm/kg) of the initial osmolality (412 and 398 mOsm/kg respectively for the atropine solution with and without preservatives) after 6 months of storage at 25 °C. Moreover, pH did not vary by more than 1.8% (0.1 pH unity) of the initial pH (6.1 for both solutions).

For all studied conditions, mean atropine concentrations did not vary by more than 5.7% of mean initial concentrations (as presented in Figure 4). By extrapolation of the degradation rate using a linear regression, it could be estimated that atropine concentrations would remain higher than 90% of the original concentration for about 300 days.

Chromatographs showed no sign of BPs until day 8 for both types of LDPE eyedroppers at 25 °C. After 15 days of storage, one BP appeared, presenting a retention time of 2.10 min (relative retention time of 0.2; Figure 5A) seemingly not detected during forced degradation assays but close to that of tropic acid (Figure 5B). However, when diluting the know impurity tropic acid in a phosphate buffer of the same nature and concentration of the one used for the atropine formulation, its retention time changed to be identical to that of the breakdown product (Figure 5C), thus indicating that it is highly likely that the misidentified breakdown product is in fact tropic acid.

Figure 6 shows the increase of tropic acid concentrations during the 6 months of the study. After 6 months, tropic acid concentrations were 3.5 and 3.4 µg/mL, respectively, for both the formulation with and without the preservative.

#### 3.2.3. Sterility Assay

None of the five analysed solutions conserved in unopened bottles at day 0, day 60, and day 180 showed any signs of microbial growth.

### 3.3. Atropine Concentrations in Eye Drops During Simulated Use

During 6 days of drop sampling, no variation exceeding ±5% of initial concentration was found for any of the studied conditioned as presented in Figure 7.

## 4. Discussion

Our study presents new data on the physicochemical stability of 2 formulations (with and without antimicrobial preservatives) of a 0.1 mg/mL atropine solution conditioned in two differently sterilized LDPE eyedroppers (gamma radiations and ethylene oxide) which can be used to prevent myopia progression in children.

The method used for the quantification of atropine was the one described in the European Pharmacopeia, atropine monography, and the use of atropine impurities allowed the identification of 2 breakdown products that appeared during forced degradation to validate the method as being stability-indicating: tropic acid (impurity C) and apoatropine (impurity A). These two compounds have already been widely described as being the main instability by-products occurring during either hydrolysis (tropic acid and tropine) or dehydration (apoatropine) [39]. Our forced degradation results are coherent with previously described data suggesting higher stability in acidic conditions and lower half-life in neutral (pH 7–8) or alkaline (pH > 8) conditions [40]. Using the same chromatographic method, we were also able to quantify tropic acid, which presented a retention time of 2.5 min in the absence of a phosphate buffer. However, in the presence of the phosphate buffer used in the formulations, its retention time was reduced to about 2 min. This could be explained by a local modification of pH induced by the buffer (pH of the formulation = 6.1) leading to tropic acid being deprotonated (the pKa of tropic acid being of 4.3 [41]) and thus being less retained by the stationary C18 column than when injected without the buffer, as the pH will therefore be imposed by the mobile phase (pH 3.9) and tropic acid will be in unionized protonated form. The choice of the quantification wavelength at 210 assured maximum sensitivity with minor influence of the mobile phase, and the absence of dilution of the formulations before chromatographic analysis also meant that we were able to detect minute quantities of breakdown products in the ophthalmic solutions.

For both of the tested formulations, all parameters were in favour of a physicochemical stability of 6 months. No modifications of visual aspect, chromaticity, or luminance were detected. Chromaticity analyses would have helped to detect minute modifications of colour [42], invisible to the naked eye, but as such, the solutions remained colourless throughout the study. Turbidity analyses did not reveal the formation of any additional particles, and the pH and osmolality values also remained within specifications and were compatible with an ophthalmic administration route. Atropine concentrations also remained within specifications after 6 months of storage (maximum loss of 5.3% of initial mean concentration), thus conserving good therapeutic efficiency during the conservation period. However, they did decrease for both formulations following a linear degradation rate, leading to increasing concentrations of tropic acid which reached a maximum of 3.5 µg/mL after 6 months of storage. Comparatively, the concentrations of tropic acid found in commercialised atropine ophthalmic specialities were 6 to 11 times higher (21.4 and 40.2 µg/mL versus 3.5 µg/mL) (see Appendix A). This information is therefore in favour of a limited clinical impact of the appearance of tropic acid in the formulations during storage, even if studies evaluating the possible toxicity of tropic acid during chronic use could be performed to verify this point. By extrapolation of the evolution of measured atropine concentration over time, assuming it will follow in the worst case a zero-order reaction [43], it could be hypothesised that a 10% decrease would be reached within 300 to 350 days. As the degradation of atropine is linked to the hydrolysis of the ester function in an aqueous media, the equilibrium state will likely not be reached immediately, therefore making this hypothesis plausible, but it would however need to verify if precise data past 6 months is needed. Also, if longer stabilities are one day required, refrigerated conservation conditions could be tested (as lower temperatures will lead a slower breakdown rate by slowing down atropine hydrolysis), or alternatively, the pH of the formulation could be reduced to reach the maximum stability range of atropine (pH comprised between 3 and 4) [44,45]. However, this could decrease ophthalmic tolerance, especially for paediatric patients, which is the target population for which these formulations are designed. In the formulations we tested, the pH was buffered to a more physiological value (around pH 6), thus hopefully limiting potential unpleasantness for the children during administration and therefore increasing therefore adherence to treatment as well as suppressing any potential modifications of pH that have already been reported during the use of gamma-sterilized LDPE eyedroppers [28].

The sterility assay that we performed in our work followed the European Pharmacopeia sterility monography [46] and, in our case (for our preparation conditions), did not reveal any microbial contamination in units stored for up to 6 months. Microbiological testing of the final product is an essential test to be able to assign a microbiological shelf-life (along with prerequisites such as controlled cleanroom environment, validated preparation conditions, and integrity testing of the container and closure [47]). Concerning the container and closure system, the gamma-sterilized eyedroppers were provided with a closure system (Novelia^®^ caps) which does not allow unfiltered air to penetrate the eyedropper, is commercially guaranteed to preserve sterility of the content more than a month [48], as well as has been tested for various hospital-prepared ophthalmic solutions either after one month of simulated use or after freezing and thawing [27,29].

During the simulated use study, atropine was quantified in the emitted drops to evaluate any potential loss of API by sorption (adsorption or absorption), as it has been shown that such a phenomena can occur during the first few days of treatment initiation with certain APIs and ophthalmic delivery devices [49]. During the 6-day study, atropine concentrations in the emitted drops remained stable (no variation exceeding of ±5% of initial concentration), therefore excluding any clinically significant loss of atropine. Such information is of high clinical value, as it correlates directly with the drop volume to determine the quantity received by the patient and needs to be known by prescribers. It also indicates that the preservative-free formulation can be used safely with the sterility preserving device that was tested (using Novelia^®^ caps). As antimicrobial preservatives are known to cause potential side effects, especially during long-term treatments [50,51,52], this information is also of high clinical value.

The stability of atropine in aqueous solutions has been studied over the years, mainly for injectable usage and at concentrations higher than 1 mg/mL, but there is little pertinent data regarding the stability of more diluted forms. Driver et al. followed atropine concentrations of 0.4 mg/mL atropine sulphate solution stored at room temperature for up to 8 days and did not notice any change in concentrations [24], but no other physicochemical parameters were evaluated and the exhaustive composition of the formulation was not detailed. More recently, the physicochemical stability of a 0.1 mg/mL atropine sulphate solution prepared by diluting 10 mg/mL commercial atropine single doses in 0.9% sodium chloride was evaluated for 6 months at room temperature [25]. The authors reported less than 5% loss of atropine and did not mention any breakdown product detection, but the wavelength of their chromatographic method was set up at 250 nm, which is not the wavelength recommended by the European Pharmacopoeia for the analysis of atropine (210 nm) and is less advantageous for the detection of breakdown products in minute quantities. Saito et al. also reported a physicochemical stability of 6 months of 1% commercial atropine solutions diluted in 0.9% saline solution to concentrations of 0.1 mg/mL as well as higher concentrations ranging up to 5 mg/mL [26]. Despite using a tandem mass spectrometry liquid chromatography method initially developed for the quantification of atropine in plasma for clinical and forensic purposes [53], the authors did not mention the presence specifically of tropic acid or of any other breakdown products, possibly because the method they used was not validated as stability-indicating or because breakdown products were not even researched, as the research or quantification of such products was not included in their acceptability criteria.

Stability studies are of vital importance to be able to assign a shelf-life to medications; however, they must be conducted properly to avoid any misinterpretation or incomplete conclusions [54]. Several guidelines exist, such as those prepared by the International Conference on Harmonisation for stability studies [37], by the US Pharmacopoeia [55], or by international consensus by learned societies [38,56,57]. In our case, we validated a European Pharmacopoeia monography initially designed for the quality control of API atropine as being stability-indicating and capable of detecting and identifying potential breakdown products. Our results are concordant with previously published data concerning the stability of low concentration atropine solutions with regards to the concentration of atropine; however, we showed that atropine degradation is present and does lead to increased concentrations of tropic acid, a known breakdown product. Very little information is available about the toxicity of tropic acid and even less is available about any potential ocular toxicity. In the human body, atropine metabolism produces small (3%) amounts of tropic acid [58] and atropine is mostly excreted unchanged in the urines. During ophthalmic administrations in rabbits, it has been shown that atropine has good ocular bioavailability via both transcorneal and transconjunctival-scleral routes [59]. Unfortunately and to the best of our knowledge, atropine metabolism in the eye has not been studied despite there being a clear incentive to do so, like for other drugs [60]. There is therefore very little information available about the possibility of the natural formation of tropic acid in the eye after atropine topical application or, as stated previously, about tropic acid’s ophthalmic toxicity. In this work, we found that concentrations of tropic acid reached 3.5 µg/mL in the atropine formulations we tested, those concentrations being 6 to 11 times lower than the quantities we also detected in commercial atropine ophthalmic formulations available on the marked. As such, it could be considered safe to administer the tested formulations after 6 months of conservation.

## 5. Conclusions

This study brings new information about the behaviour and stability of low-dose atropine ophthalmic solutions, showing that 0.1 mg/mL atropine solutions buffered at a physiological pH, with or without antimicrobial preservative, are physiochemically stable for 6 months at 25 °C when stored in LDPE bottles, with an identical microbial shelf-life.

## Figures and Tables

**Figure 1 pharmaceutics-12-00781-f001:**
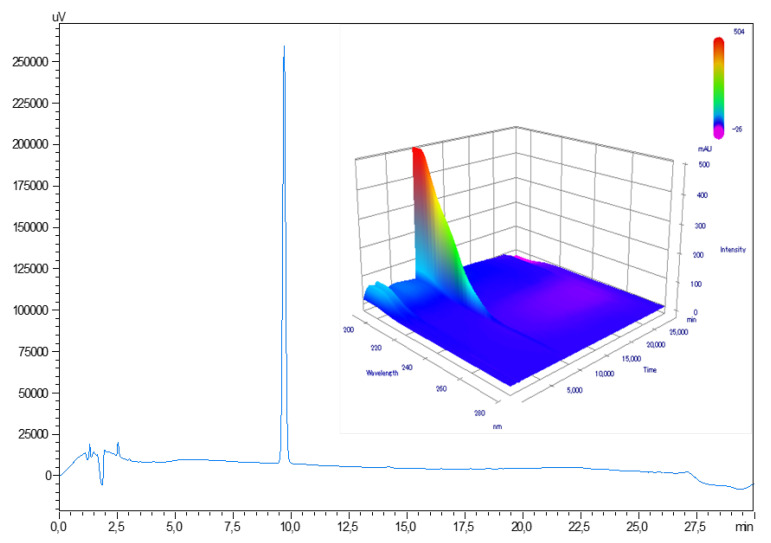
Reference chromatogram of a 0.1 mg/mL atropine solution at 210 nm and with diode array detector screening.

**Figure 2 pharmaceutics-12-00781-f002:**
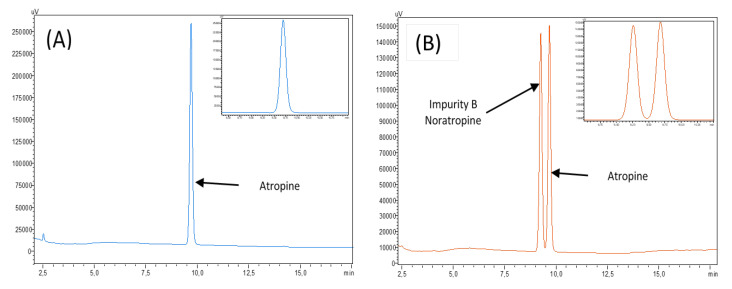
Chromatograms of atropine impurities at 210 nm: (**A**) reference atropine chromatogram, (**B**) chromatogram of the atropine impurity B CRS solution, (**C**) chromatogram of atropine for the peak identification CRS (containing impurities A, D, E, F, G, and H) solution, and (**D**) chromatogram of the tropic acid R (impurity C) solution. The insets represent a close up of the chromatograms.

**Figure 3 pharmaceutics-12-00781-f003:**
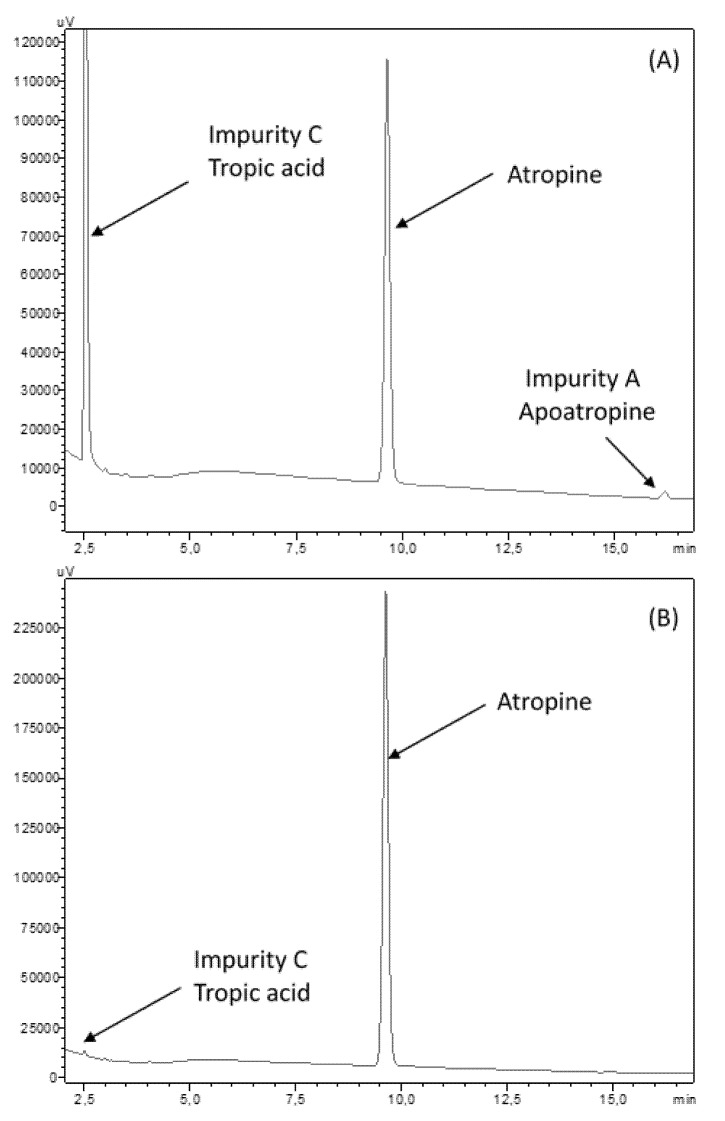
Chromatograms at 210 nm of breakdown products (BPs) obtained after forced degradation: (**A**) alkaline conditions of NaOH 0.5 N for 0.5 h and (**B**) acid conditions of HCl 0.1 N for 1 h at 90 °C.

**Figure 4 pharmaceutics-12-00781-f004:**
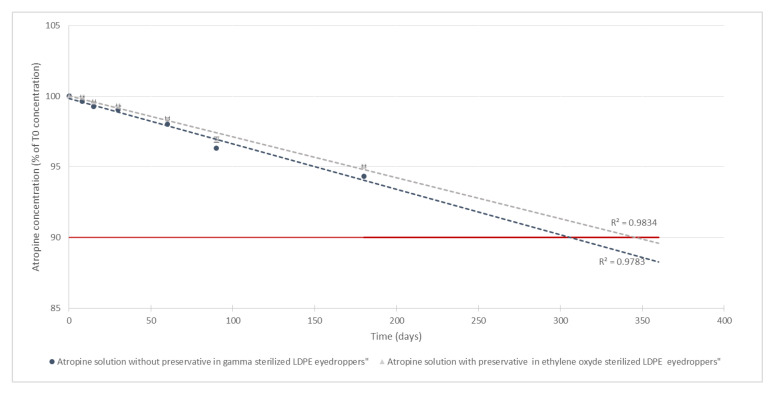
Evolution of atropine concentration over time (*n* = 5, mean ± 95% confidence interval): the grey curve is the atropine solution with preservatives conditioned in low-density polyethylene (LDPE) CAT^®^ eyedroppers. The blue curve is the atropine solution without preservatives conditioned in LDPE NOVELIA^®^ eyedroppers. The red line is 90% of the initial concentration. The dotted lines are the calculated linear regression of the evolution of atropine concentrations.

**Figure 5 pharmaceutics-12-00781-f005:**
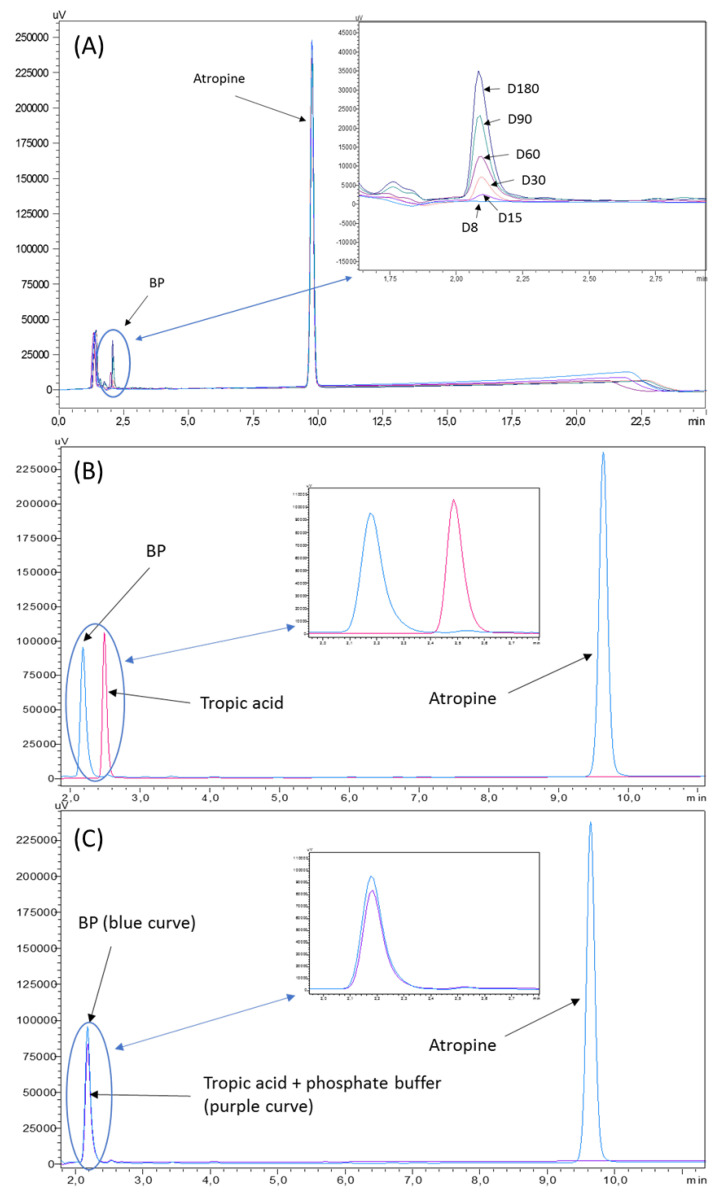
Chromatograms of (**A**) the increased area under curve of the breakdown product, (**B**) superposition of chromatograms of tropic acid diluted in water and an atropine solution containing the breakdown product, and (**C**) superposition of chromatograms of tropic acid diluted in phosphate buffer and an atropine solution containing the breakdown product.

**Figure 6 pharmaceutics-12-00781-f006:**
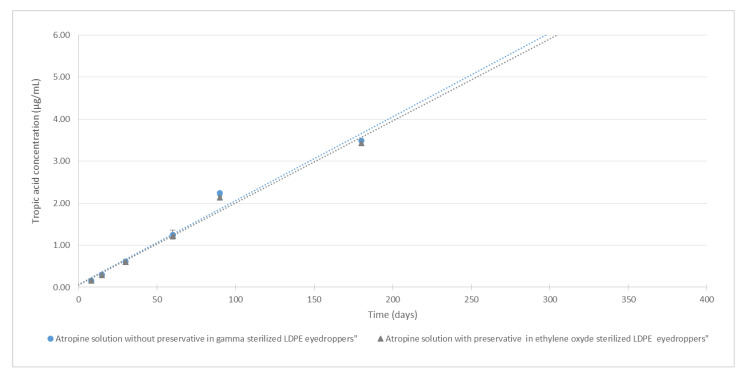
Evolution of breakdown product (tropic acid) concentration during time (*n* = 5, mean ± 95% confidence interval): The grey curve is the atropine solution with preservative conditioned in LDPE CAT^®^ eyedroppers. The blue curve is the atropine solution without preservative conditioned LDPE NOVELIA^®^ eyedroppers. The dotted lines are the linear regression of the evolution of breakdown product concentrations.

**Figure 7 pharmaceutics-12-00781-f007:**
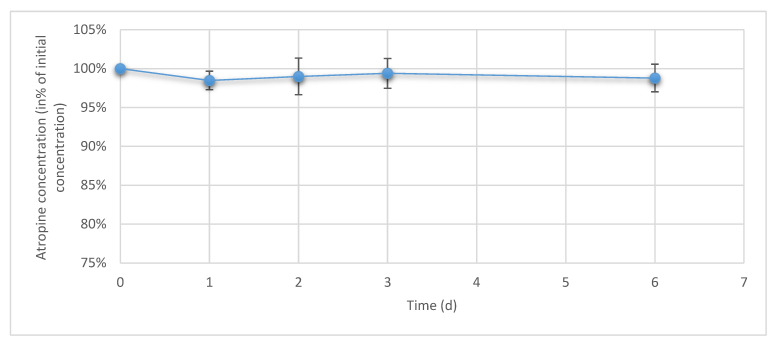
Evolution of atropine concentration in the emitted drops during simulated use (*n* = 3 times 10 pooled drops, mean ± 95% confidence interval).

**Table 1 pharmaceutics-12-00781-t001:** Composition of the tested atropine ophthalmic formulations. q.s: quantity sufficient.

Chemical Components	Formulation (mg)	
	Without Preservative	With Preservative
Atropine sulphate (batch 18276508, exp. 31/01/2021, Inresa, France)	100	100
Natrium dihydrogenophosphate dihydrate (NaH_2_PO_4_) (batch 190298040, exp. 30/11/2021, Inresa, France)	7800	7800
Dinatrium monohydrogenophosphate dodecahydrate (Na_2_HPO_4_) (batch 18129611, exp. 30/04/2023, Inresa, France)	4480	4480
Cetrimide (batch 16F08-B01-334049, exp. 05/2020, Fagron, Netherlands)		100
Sodium chloride (NaCl) 0.9% (Versylene^®^; Fresenius Kabi France, Louviers, France)	q.s 1000 mL	q.s 1000 mL

**Table 2 pharmaceutics-12-00781-t002:** Gradient used for the liquid chromatography (LC) mobile phase.

Time (min)	Mobile Phase (%)	
	A	B
0	95	5
2	95	5
20	70	30
21	95	5
25	95	5

**Table 3 pharmaceutics-12-00781-t003:** Atropine impurities retention times and relative retention times.

Impurity Retention Times		
	Experimental Absolute Retention Time (min)	Relative Retention Time
Atropine	9.7	1
Impurity A	16.2	1.7
Impurity B	9.3	0.9
Impurity C	2.5	0.3
Impurity D	7.6	0.8
Impurity E	7.1	0.7
Impurity F	8.0	0.8
Impurity G	10.8	1.1
Impurity H	9.3	0.9

**Table 4 pharmaceutics-12-00781-t004:** Evolution of turbidity over time. *n* = 1 (pooled volume from 5 units). FNU: Formazin Nephelometric Units.

	Turbidity (FNU)						
	Day 0	Day 8	Day 15	Day 30	Day 60	Day 90	Day 180
Atropine solution with preservative conditioned in LDPE CAT^®^ eyedroppers	0.33	0.31	0.27	0.78	0.78	0.43	0.93
Atropine solution without preservative conditioned LDPE NOVELIA^®^ eyedroppers	0.32	0.31	0.26	0.54	0.44	0.34	0.64

**Table 5 pharmaceutics-12-00781-t005:** Evolution of pH and osmolality over time (*n* = 5, mean ± 95% confidence interval).

		Day 0	Day 8	Day 15	Day 30	Day 60	Day 90	Day 180
Atropine solution with preservative conditioned in LDPE CAT^®^ eyedroppers	pH	6.10 ± 0.01	6.11 ± 0.01	6.13 ± 0.02	6.13 ± 0.01	6.13 ± 0.02	6.21 ± 0.04	6.12 ± 0.01
	Osmolality (mOsm/kg)	412 ± 16	400 ± 6	403 ± 14	393 ± 14	400 ± 5	413 ± 11	418 ± 23
Atropine solution without preservative conditioned LDPE NOVELIA^®^ eyedroppers	pH	6.10 ± 0.01	6.11 ± 0.01	6.13 ± 0.01	6.14 ± 0.02	6.13 ± 0.01	6.21 ± 0.01	6.09 ± 0.01
	Osmolality (mOsm/kg)	399 ± 2	401 ± 6	409 ± 6	405 ± 2	415 ± 15	408 ± 10	405 ± 7

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
