# Peer review of "Stability of Ophthalmic Atropine Solutions for Child Myopia Control"

_pharmaceutics, 2020, doi:10.3390/pharmaceutics12080781_

Round 1

Reviewer 1 Report

The experimental design was well organized, the methods are described in detail and clearly. The stability of the solutions with and without preservative packaged in multidose bottles has been verified both by means of visual inspections and by means of chromatic analyzes and by means of quantitative analyzes of the active principle, checking for the presence of degradation products. The analytical method made it possible to detect even small quantities of degradation products such as tropic acid. It is not so surprising that the degradation products are reduced compared to when the drug is used in higher concentrations and in any case the presence of tropic acid, even if in small quantities, suggests a possible toxicity in chronic use.

Author Response

We thank reviewer for his positive comments. A mention that studies evaluating the possible toxicity of tropic acid during chronic could be performed to verify this point was added to the discussion, lines 410 to 412.

Reviewer 2 Report

Line 13-14: how is the expiration date determined in commercial products if no information on the drug stability?

How were the preservative chosen? Are they commonly used?

Is LDPE a common container material for the marketed drug solution?

Why only 0.1 mg/mL concentration? Is the purpose to analyze the stability at 0.1 mg/mL for LDPE long-term storage container?

Author needs to cite any other stability studies in this regard and how they differ from their’s and value over the other stability data published till now.

Line 86: specify the preservative used and why (in materials or in discussion).

Explain why two different types of containers. What was the size of the container and solution volume in each container.

What temperature, mixing speed, batch size was the solution made in? Materials and methods sections is very brief and missing details.

How was the stability maintained during formulation?

What were the simulated in-use conditions?

Figure 2: why different impurities in different figures? Cant they all be identified in one chromatogram?

Line 269: What is a BP? It is introduced for the first time and not defined. I suppose it means breakdown products? Needs more explanation.

There is no mention of forced degradation in methods and materials and the purpose/ selection of conditions for forced degradation is also not discussed.

Section 3.2.1: need to show the turbidity data over 6 months.

Section 3.2.2: show the actual osmolality data, same for pH.

Author Response

Comments and Suggestions for Authors

Line 13-14: how is the expiration date determined in commercial products if no information on the drug stability?

Answer: The sentence of the abstract was clarified, and now reads: However there are currently no commercial industrial forms available and there is little data published concerning the stability of medications prepared by compounding pharmacies. See lines 13 and 14.

How were the preservative chosen? Are they commonly used?

Answer: Cetrimide was chosen because it is a preservative commonly used for the antimicrobial preservation of ophthalmic solutions, with an efficacy similar to benzalkonium chloride (see Velpandian, T. Preservatives for Topical Ocular Drug Formulations. In Pharmacology of Ocular Therapeutics; Ed.; Springer International Publishing: Cham, 2016; pp. 419–430 ISBN 978-3-319-25498-2, and Dao et al. Microbial Stability of Pharmaceutical and Cosmetic Products. AAPS PharmSciTech 2018, 19, 60–78, doi:10.1208/s12249-017-0875-1.). This information as added lines 108 to 110.

Is LDPE a common container material for the marketed drug solution?

Answer: LDPE is a very common material used for conditioning of ophthalmic solutions, as most recent containers use this material and have been used in published stability studies. This affirmation and adequate references have been added to the introduction section line 77 and 78.

Why only 0.1 mg/mL concentration? Is the purpose to analyze the stability at 0.1 mg/mL for LDPE long-term storage container?

Answer: As stated in the introduction section, lines 77 to 79, the aim of this study was to assess the physicochemical stability and to control the sterility of two 0.1 mg/mL atropine ophthalmic in two different low-density polyethylene multidose eyedroppers. The 0.1 mg/mL concentration was chosen because it is the concentration with currently the most data concerning safety and efficacy. This information was added to the manuscript line 80.

Author needs to cite any other stability studies in this regard and how they differ from their’s and value over the other stability data published till now.

Answer: An in-depth analysis and discussion of previously published stability data concerning ophthalmic atropine solutions is presented in the discussion section, lines 448 to 467. However, some of the shortcomings of such studies have now been indicated in the introduction section, lines 74 to 76.

Line 86: specify the preservative used and why (in materials or in discussion).

Answer: As specified in Table 1, cetrimide was the preservative used. This information was added to the text of the manuscript line 92. Cetrimide was chosen because it is a preservative commonly used for the antimicrobial preservation of ophthalmic solutions, with an efficacy similar to benzalkonium chloride (see Velpandian, T. Preservatives for Topical Ocular Drug Formulations. In Pharmacology of Ocular Therapeutics; Ed.; Springer International Publishing: Cham, 2016; pp. 419–430 ISBN 978-3-319-25498-2, and Dao et al. Microbial Stability of Pharmaceutical and Cosmetic Products. AAPS PharmSciTech 2018, 19, 60–78, doi:10.1208/s12249-017-0875-1.). This information as added lines 108 to 110.

Explain why two different types of containers. What was the size of the container and solution volume in each container.

Answer: Single-dose container technology is not readily available to compounding pharmacies, and multidose eyedroppers are therefore the most used container. However not all of those devices possess a system allowing their content to be preservative-free, and their contents must therefore be preserved. In this work, two different low-density polyethylene (LDPE) multidose eyedropper (one with a sterility preserving technology, the other without) were chosen so has to allow a choice of container depending on stability data. The introduction section was modified to accordingly, lines 83 to 85. The maximum filling capacity of the eyedroppers was of 8 mL and was added line 113 after the information already present about the solution volume inserted into them (6 mL).

What temperature, mixing speed, batch size was the solution made in? Materials and methods sections is very brief and missing details.

Answer: The formulations were prepared by dissolving atropine into the 0.9% sodium chloride solution, at room temperature under gentle agitation, before adding the hydrogenophosphate buffer, and finally, if needed, the preservative(cetrimide). For the puropose of the study, batch size was of 1L, for both formulations. This information was added to the materials and methods section, lines 105 to 108.

How was the stability maintained during formulation?

Answer: Initial day 0 analyses were performed immediately after conditioning, within 4 hours after the end of the preparation of the solutions, to have results as representative as possible of initial conditions (least degradation or modification of parameters). This information was added to the manuscript lines 130 to 132.

What were the simulated in-use conditions?

Answer: In order to simulate normal patient use, every day for 6 days, one drop from each eyedroppers was manually emitted (ie. the drop was squeezed out of the bottle as if to be administered to the eye, but instead of being administered it was collected for analysis), at room temperature. This information was added to the manuscript lines 135 to 137.

Figure 2: why different impurities in different figures? Cant they all be identified in one chromatogram?

Answer: The chromatograms presented in Figure 2 show separately different impurities as they come from different European Pharmacopoeia reference solutions, and were thus analyzed sequentially, in order to be able to correctly identify each peak using the relative retention times provided in the atropine monography. This information was added to the manuscript line 274 to 277.

Line 269: What is a BP? It is introduced for the first time and not defined. I suppose it means breakdown products? Needs more explanation.

Answer: Breakdown products (BP) are first mentioned line 127 of the manuscript. In order to clary, it has now been specified in the manuscript that breakdown products research means specifically looking for products resulting from the degradation of atropine (see lines 127-128).

There is no mention of forced degradation in methods and materials and the purpose/ selection of conditions for forced degradation is also not discussed.

Answer: Forced degradation conditions are explained lines 200 to 206 in the manuscript. In order to help make this paragraph stand out, an extra line was skipped before and after the paragraph.

Section 3.2.1: need to show the turbidity data over 6 months.

Answer: Complete turbidity data has now been presented in Table 4 (lines 310 to 313)

Section 3.2.2: show the actual osmolality data, same for pH.

Answer: Complete pH and osmolality data has now been presented in Table 5 (lines 322 and 324)

Reviewer 3 Report

This manuscript investigates stability of ophthalmic atropine solutions. The objective of this study was to evaluate the stability of two 0.1 mg/mL atropine formulations (with and without antimicrobiobial preservative) for 6 months, in two different low-density polyethylene (LDPE) multidose eyedroppers.

The manuscript is well structured, but the novelty is questionable.

Here are some of comments and questions

  • Please detail the stability issues of atropine in eye drops in the introduction section
  • line 70-77 need references
  • Please standardize the following term in the manuscript: multi dose eyedroppers or multidose eyedroppers
  • line 111 need ICH guideline reference
  • Please modify the unit mosm/kg to mOsm/kg
  • Figure 6 appears twice in the article, please corrected, and the Figure 6 is not mention in the text, please corrected
  • Please improve the Conclusion section in order to emphasize the novelty of the paper

Author Response

Here are some of comments and questions

Please detail the stability issues of atropine in eye drops in the introduction section

Answer: An in depth analysis and discussion of previously published stability data concerning ophthalmic atropine solutions is presented in the discussion section, lines 447 to 466. However, some of the shortcomings of such studies have now been indicated in the introduction section, lines 74 to 76.

line 70-77 need references

Answer: Adequate references have been added to this paragraph, see lines 74 to 78.

Please standardize the following term in the manuscript: multi dose eyedroppers or multidose eyedroppers

Answer: The term “multidose” has been applied throughout the manuscript (see line 93 for the change made).

line 111 need ICH guideline reference

Answer: ICH guideline Q1B compliancy added to the climate chamber, line 122.

Please modify the unit mosm/kg to mOsm/kg

Answer: The unit mOsm/kg has been applied throught the manuscript (see lines 317).

Figure 6 appears twice in the article, please corrected, and the Figure 6 is not mention in the text, please corrected

Answer: In the uploaded and subsequently downloaded version of the manuscript, Figure 6 appears once in the manuscript and is mentioned line 351.

Please improve the Conclusion section in order to emphasize the novelty of the paper

Answer: The conclusion was enhanced to emphasize the novelty of the paper, and now reads:

“This study brings new information about the behaviour and stability of low dose atropine ophthalmic solutions, showing that 0.1 mg/mL atropine solutions buffered at a physiological pH, with or without antimicrobial preservative, are physicochemically stable for 6 months at 25°C when stored in LDPE bottles, with an identical microbial shelf-life”. See lines 493 to 496.